# The Translated and Adapted Brazilian Version of the Behavioral Enabling Scale for Family Members of Psychoactive Substance Users: An Analysis of the Factorial Structure and Internal Consistency

**DOI:** 10.3390/ijerph21091230

**Published:** 2024-09-18

**Authors:** Heloisa Praça Baptista, Hilda Maria Rodrigues Moleda Constant, Cassandra Borges Bortolon, Helena Maria Tannhauser Barros

**Affiliations:** 1Department of Toxicology and Pharmacology Postgraduate Program in Health Sciences, Federal University of Health Sciences of Porto Alegre, Porto Alegre 90050-170, RS, Brazil; psico.heloisabaptista@gmail.com (H.P.B.); helenbar@ufcspa.edu.br (H.M.T.B.); 2Department of Behavioral Sciences Methodology, Faculty of Psychology and Speech Therapy, University of Valencia, 46010 Valencia, Valencian Community, Spain; 3Department of Psychiatry, Postgraduate Program in Psychiatry and Medical Psychology, Federal University of São Paulo, São Paulo 04021-001, SP, Brazil

**Keywords:** family relationships, enabling, psychometry, substance-related disorders

## Abstract

Objective: The enabling behaviors of family members of psychoactive substance users can be crucial in maintaining addiction. There are no psychometrically evaluated instruments to measure the frequency of the enabling behaviors of family members of psychoactive substance users. Therefore, this study aimed to assess the internal consistency and factor structure of the Behavioral Enabling Scale. Design: A cross-sectional study was carried out, with a secondary analysis of data collected from 400 family members of psychoactive substance users that used a hotline service in Brazil. Exploratory factor analysis was conducted with an initial sample of 200 protocols, and with the remaining 200 protocols, a confirmatory factor analysis was conducted. Results: The internal consistency estimate proved entirely satisfactory in both samples, where the first presented a Cronbach’s alpha of 0.76 and the second had a consistency estimate of 0.79. Factor analysis was conducted using a shortened version of the instrument, with 15 items, during which six factors that cover 65% of the scale’s explained variance were extracted. KMO = 0.68 and Bartlett’s test of sphericity = X^2^ (gl = 153) 497.201, *p* < 0.0001, were significant. Conclusion: The Brazilian version of the Behavioral Enabling Scale is a valid tool that measures the frequency of the enabling behaviors of family members of psychoactive substance users. The measurement instrument enables further investigations into the behavior of family members regarding the use of psychoactive substances by their relatives.

## 1. Introduction

The use of psychoactive substances significantly damages the lives of drug users and their family systems. Orford summarized the core of affected family members’ experiences and reported that family members of drug users most often experience concern for their relatives, family disharmony, exposure to threats, a lack of accurate information, coping with several dilemmas, high levels of stress and impaired health, and a lack of adequate healthcare [1,2,3,4]. The experience of affected family members (AFMs) with the use of psychoactive substances by an adult family member is described with complaints such as concern about the user’s health and professional, social, and family performance; the precarious health condition of AFMs, who suffer physical and psychological damage; sleep disturbances; increased use of substances such as alcohol, tobacco, and medications; and destabilization of family members’ relationships. In addition, feelings of anxiety, sadness, anger, fear, guilt, shame, helplessness, and hopelessness are similarly noted in different cultures [1,5,6]. In previous studies, we reported that the family members were mainly women; mothers or spouses; homemakers; those with low incomes; and those with less than eight years of education [2,4]. Some complicating factors can increase the burden experienced by AFMs. The overload of tasks, the personal and domestic financial demands related to the user, the stigma of having a user in the family, and the lack of social support for AFMs make the experience of having a user in the family even more intense [1,6,7].

The family system is considered an important factor in the development of individuals. How the system is organized, the construction and maintenance of bonds, and communication between members can be considered protective factors or risk factors for psychological disorders. In cases of psychoactive substance use disorder, the behavior of family members can be an important agent in stopping psychoactive substance use, thus serving as a support network and an important protection factor against relapses. However, the behavior of family members can also facilitate use and, thus, become a risk factor for relapses [8]. Thus, the problematic use of psychoactive substances demands care and treatment beyond the user, as it is not an individual problem, but rather a problem involving the family system [9].

The coping behaviors adopted by AFMs are crucial in developing and maintaining addiction. Enabling behaviors are characterized as a group of behaviors developed by AFMs as a coping strategy for stress related to substance use by a family member. However, these behaviors can be unintentionally considered reinforcers or facilitators of continued use [9]. These behaviors can influence the user’s substance consumption through positive or negative reinforcement, increasing the likelihood of future use. Mexican–American heroin users perceived that they received support regarding substance use, either directly or indirectly, from their family members [10]. These and other data [9,11] demonstrate the role of the family system in the initiation and continuation of problematic substance use by enabling the drug use of a relative. Behaviors such as providing money to buy the drug, covering up the truth about the use, making excuses to other people about the user’s behavior, using psychoactive substances together with the user, covering up or taking the blame in the user’s place, making threats without the intention to go through with them, helping the user to recover from the effects of an acute hangover, and maintaining household routines as if nothing is happening are some of the enabling behaviors performed by AFMs [9,10,12].

To assess the presence and frequency of these behaviors, Rotunda, R.J.; West, L.; and O’Farrell, T.J. [13] created the Behavioral Enabling Scale (BES), which was applied to the partners of alcohol users and the users themselves who participated in the Counseling for Alcoholics’ Marriages Project (Project CALM), a behavioral follow-up program developed at Harvard Medical School, USA, to assess the frequency of behaviors that could maintain or even be partially responsible for increasing users’ alcohol consumption. The results showed that their partners demonstrated at least one form of enabling behavior during the year before the study. The most prevalent behaviors were taking over the user’s tasks, lying to cover up their use, and even using the substance together.

This study is not intended to label, stigmatize, blame, or pathologize AFMs, but rather to identify the frequency of the enabling behaviors that are part of maladaptive coping strategies, aiming to reduce the impact caused by substance use on AFMs [1,7,13]. Coping strategies involving enabling behaviors require AFMs to focus intensely on the user to minimize the damage caused by psychoactive substance use. Adaptive coping strategies are more related to the removal of AFMs from the role of caregiver of the user, with a focus on promoting physical and psychological well-being by introducing self-care behaviors to the AFMs themselves [13]. Therefore, this study aimed to analyze the psychometric properties of the BES in Brazilian Affected Family Members who sought help through a hotline service. To date, no studies have performed psychometric analyses of the BES in any culture. The BES (Brazilian version) is an adaptation of the Behavioral Enabling Scale developed by Rotunda, R.J.; West, L.; and O’Farrell, T.J. (2004) [14]; in addition to translation, the scale was also adapted to expand its use to other family members in addition to spouses.

## 2. Methods

### 2.1. Design

A cross-sectional study was carried out involving the secondary analysis of data already collected from family members of users of psychoactive substances through a hotline service. This report refers to two sets of data from two stages of the same database analysis: Exploratory Factor Analysis (EFA) and Confirmatory Factor Analysis (CFA). Factor analysis was conducted via the modified scale, with no questions about the spouses.

### 2.2. Participants

We analyzed 400 protocols regarding family members of psychoactive substance users who called the Ligue 132 hotline service of the National Service for Guidance and Information on Drug Use. Ligue 132 is a hotline service that provides counseling and information about psychoactive substances, free of charge and fully confidential, operating through telephone calls, serving the five regions of Brazil. The protocols analyzed in the present study were generated from January 2015 to December 2016. The consultations were performed by undergraduate health students previously trained in neuroscience for treating and preventing psychoactive substance use. All consultations were supervised by postgraduate students in the health area and researchers, who, in turn, were supervised by professors and senior researchers. More details of this process can be found in our previously published work [3].

Our study’s inclusion criteria were stringent, ensuring the quality and relevance of the data. We selected protocols from family members over 18 years of age who had completed their first consultation and agreed to participate in the study at the time of consultation. This agreement was obtained through the informed consent form, which was completed verbally during the proactive telephone call initiated by the family member. We excluded protocols from family members of tobacco-only users and those with incomplete first-care protocols. The study was ethically approved by the Ethics Committee of the Universidade Federal de Ciências da Saúde de Porto Alegre (UFCSPA, CEP no. 4.217.827).

### 2.3. Instruments

#### 2.3.1. Sociodemographic Questionnaire

A standard service questionnaire for all the people who called the “Ligue 132” hotline presented questions regarding age, gender, marital status, educational level, and family income.

#### 2.3.2. Family Care Protocol

An exclusive service questionnaire for family members of psychoactive substance users collected information on the degree of kinship, gender, age, and substance used. It also highlighted whether the family member had previously sought help to address the losses resulting from the use of substances by their family member and whether they used psychoactive substances.

#### 2.3.3. Behavioral Enabling Scale—BES (Brazilian Version)

A translation and adaptation of the BES was developed [13] to assess the enabling behaviors of the spouses of alcohol users. The author authorized the use of the instrument and the adaptation that would allow it to be applied to other family members in addition to spouses. The original scale is composed of 20 items that assess the frequency of enabling behaviors on a Likert scale ranging from “never” to “very often” (1 = never, 2 = rarely, 3 = sometimes, 4 = often, and 5 = very often). The instrument assesses the period of the last 12 months. To expand the applicability of the scale to family members in general (spouses, partners, parents, and siblings, among other significant relationships), two items specific to spouses were removed. This adapted version of the instrument was also extended for application to family members of users of other psychoactive substances in addition to alcohol, marijuana, cocaine, crack, and inhalants.

The translation process, cross-cultural adaptation, and validation of the BES content followed the procedures described by Constant et al. [15]. Briefly, the BES was translated from the original instrument (English version) to Portuguese by a Brazilian researcher with proficiency in English, and the translated version (Portuguese) was back-translated into English by a native English-speaking person with fluency in Portuguese. The re-translator did not participate in the first step and was not familiar with the original instrument. The translation consolidation was the second step, and the committee method was used to verify whether the proposal for the final version was close to the original version. The objective was to unify the versions of the translations/retransmissions and reach a consensus regarding the content explored. The committee had three professionals: two Brazilian researchers with proficiency in the English language and knowledge of the field of chemical dependency (one of whom is the author of the translated version), and an English native language teacher fluent in Portuguese. The translated scale was applied to a small group of callers to the hotline service to ensure the population’s comprehension of the questions [4].

## 3. Statistical Analysis

A cross-sectional study verified the internal structure of the BES (Brazilian version) in terms of internal consistency and factor solutions.

Data were collected using an electronic spreadsheet and imported for analysis in the Statistical Package for the Social Sciences (SPSS, version 25.0) and AMOS software, version 18 [16].

A descriptive statistical analysis was performed (including measurements of central tendency and dispersion, frequency distribution, and percentage) to describe the sociodemographic data of the participants. According to the literature, the sample size was calculated using the criterion of 10 participants per scale item [17]. Since the original instrument consists of 20 items, a sample size of 200 participants was necessary. For this study, the instrument’s structure was assessed in two stages: EFA and CFA. Therefore, two independent samples were used—200 participants were used for the EFA stage, and an additional 200 participants were used for the CFA stage.

EFA was performed to investigate the factor structure of the instrument in the population studied. The extraction method with principal axis factoring was used, where the initial criterion of communality < 0.500 was adopted. To rotate the factors, an oblique quartimax rotation [18] was performed, with Kaiser normalization. CFA was performed to confirm the robustness of the factor model suggested by EFA. The Maximum Likelihood method was chosen since the sample is sufficiently large; it is a complex model with multiple factors and multiple interactions between them; and, in the presence of missing data, it works adequately. Additionally, it allows for robust statistical analysis, which benefits both this study and future research utilizing this instrument, as well as comparisons between studies. To estimate the parameters, a path diagram was created to specify the factored model. To evaluate the factorial model, adjustment indicators based on the chi-square test, root mean square error of approximation (RMSEA), comparative fit index (CFI), Tucker–Lewis index (TLI), and goodness-of-fit index (GFI) of the GFI and SRMR were used.

Finally, a bivariate analysis was carried out to investigate the power of the differences between the factors. For this analysis, the variables used were the gender of the family member, degree of kinship, family income, education, and the region of Brazil where they reside. With respect to the statistical techniques used, the analyses that involved the comparison of categorical variables between the two samples used the Pearson chi-square test. When comparisons involved continuous variables, Student’s t-test was used for independent groups. For internal consistency analysis, McDonald’s omega coefficient (ω) was used; where the results vary between 0 and 1, estimates between 0.70 and 0.95 are considered acceptable, and values of less than 0.70 may not demonstrate sufficient internal consistency for a construct [14].

## 4. Results

Information regarding the general characterization of the samples, including both family members and users in the EFA and CFA groups, is presented in Table 1. With respect to the information regarding family members, those who sought help most frequently from the Ligue 132 service for substance users were mothers, with the most representative sex being female. These family members were predominantly over 45 years old, mostly married, and had a family income of between 1 and 4 minimum wages. Regarding substance users, the majority were over 25 years old and male. Most of the investigated characteristics did not show significant differences. However, the degree of kinship (mostly mothers) and the sex of the family member (mostly female) did present differences between the EFA and CFA groups.

Initially, we checked the adequacy of the sample used for employing EFA. Two models were tested—one containing all the BES items (original version), where the KMO = 0.607, and another model where items (2, 7, and 17) were excluded and the KMO reached 0.680, indicating a better fit of the data. In addition, a screening of the number of factors to be extracted was conducted, justifying the use of a six-factor model (see scree plot in Figure 1).

Regarding the structural analysis of the scale, EFA presented satisfactory results. The Kaiser–Meyer–Olkin (KMO) measure of sampling adequacy was 0.680, and Bartlett’s test of sphericity proved to be significant (X^2^ (df = 153) 497.201, *p* < 0.000), indicating that the data were suitable for factor analysis.

During the application of EFA, we employed the principal axis factor extraction method, adopting the criterion of commonalities with values above 0.50. This careful selection process ensured that all the items were relevant to answering/explaining the variations presented by the factors (Table 2). As a result, three items (Items 2, 7, and 17 in the original scale [13]) were excluded because of their low commonality, thereby increasing the model’s explained variance. This resulted in a new version of the scale with 15 items, which is more concise.

The results obtained from EFA indicated the extraction of six factors, accounting for 60.31% of the variance explained by the scale. The scale items were distributed coherently, making the interpretation of each factor possible. Factor 1 corresponds to the “passive behavior” dimension, is composed of Items 6, 9, 10, and 13, and explains 15.73% of the variance of the scale. Factor 2 refers to the “Personal Relationships” dimension and comprises Items 3, 11, and 12, corresponding to 11.88% of the explained variance. Factor 3 represents the “Criminal Consequences of Use” dimension and comprises Items 7 and 8, representing 9.84% of the instrument’s variance. Factor 4 refers to the “Minimization of use” dimension, which includes Items 14 and 15 and encompasses 8.62% of the explained variance. Factor 5 reflects the “Active Behavior” dimension and includes Items 4 and 5, corresponding to 7.46% of the explained variance. Finally, Factor 6 is related to the “Responsibility for Use” dimension and is characterized by Items 1 and 2, representing 7.71% of the variance of the scale.

Considering the analysis that sought evidence for the structure estimated by EFA, CFA was used in a second sample, where all indices reached the desired parameters (Table 3). The practical contribution of each item to its respective factor is evidenced by the factor loadings found, where the values were greater than 0.5, reaching a maximum loading of 0.810. Most of the correlations between the factors were low (r < 0.30), which indicates that the latent variables concentrate different information from each other. The highest coefficients occurred for the “responsibility for use” factor compared with the “passive behavior” factor (r = 0.53) and “personal relationships” factor (r = 0.62) (Figure 2).

The McDonald’s omega coefficient value for the internal consistency reliability of the scale of the first sample (EFA) was ω = 0.711, demonstrating the acceptable internal agreement of the questionnaire. The six factors found after the factor analysis also had good internal reliability indices: Factor 1 “passive behavior” ω = 0.702; Factor 2 “personal relationships” ω = 0.658; Factor 3 “criminal consequences of use” ω = 0.712; Factor 4 “minimization of use” ω = 0.706; Factor 5 “active behavior” ω = 0.698; and Factor 6 “responsibility for use” ω = 0.627. The consistency estimate was not different for the second sample (CFA ω = 0.796).

## 5. Discussion

This is the first study to analyze the psychometric properties of the BES in Portuguese. Adapting the instrument to 15 items expands the use of this critical tool beyond its application to the spouses of people who abuse alcohol or illicit drugs. In clinical practice, it provides a specific and accurate assessment of different manifestations of enabling behaviors, which can be used in psychoeducation and as a first step in planning the treatment of AFMs. Treatments focusing on changing the behavior of AFMs can reduce or eliminate enabling behaviors, decrease maladaptive behaviors, and aid the development of adaptive coping strategies for difficult situations [19]. Accordingly, using the scale may help to improve family members’ behaviors toward drug users and decrease the physical and psychological suffering resulting from the stress generated by living with users [9].

The results of the instrument’s analysis demonstrate that the psychometric adaptation of the BES applied to Brazilian AFMs through a hotline has good psychometric properties. The instrument presents satisfactory reliability, and the results demonstrate good internal consistency, both in the instrument in general and in the coefficient results of each factor. We cannot perform a comparison with the results found in the original Habilitating Behavior Scale [13], which was applied to the wives of alcohol users, as it uses Cronbach’s alpha. This coefficient has certain limitations, mainly in terms of standardizing the importance of items in the model. In our study, according to the AFE model, the different items present factor loadings with different weights in the model. Therefore, we chose to use the McDonald omega to better measure reliability, as there is less risk of overestimating or underestimating the results. Furthermore, the KMO coefficient is appropriate, as the ideal value should be greater than 0.6 [20]. However, in the literature to date, there has been no exploration of the analysis of the instrument, which makes it impossible to compare these results with those of other studies. Bartlett’s test of sphericity also revealed statistical significance. The EFA indicated the extraction of six factors that covered 60.318% of the explained variance. As mentioned previously, the extraction of three items was necessary due to the low commonality of the items, which resulted in an increase in the explained variance of the model. The present study is the first in which EFA and AFC were used and the instrument was reduced, presenting a shorter version with expanded applicability for the family members of drug users in general, and not just spouses of alcohol abusers.

The interpretation of factors is based on the behaviors and phenomena evidenced in studies involving AFMs. Factor 1, “passive behavior”, describes implicit or indirect actions of AFMs that facilitate the continuation of substance use [10]. Factor 2, “Personal relationships”, presents situations related to family and social relationships in general. Disharmony in personal relationships due to substance use is a finding also described in [1,20,21,22], where AFMs chose to keep problems related to psychoactive substance use confidential (facing them in the family unit and not in public) to avoid suffering stigma for use, guilt, and shame. Factor 3, “Criminal Consequences of Use”, pinpoints the AFMs’ actions to avoid the incarceration of family members or other criminal problems related to substance use [10]. Factor 4, “Minimization of use”, suggests a normative view of substance use within the family environment [10]. Factor 5, “Active behavior”, refers to the explicit and direct behaviors of AFMs concerning use, such as using the substance with the user [10]. Factor 6, “Responsibility for Use”, refers to the behaviors of AFMs. Such behaviors have been described as “excessive responsibility”, resulting from a positive impulse by AFMs but generating negative results regarding the cessation of use [23].

Regarding sociodemographic data, the predominance of women in the sample is expected, as this is common in studies related to AFMs in different cultures [1,2,10,24]. The small number of men highlights the need for future studies focused on male AFMs. The large number of women can be understood by the culture’s socially rigid gender and identity roles, with women being seen as responsible for care in relationships, for controlling substance use, and for being more aware of the problem [23]. The ability to cope with the stress caused by the addiction of a family member makes sense if we think of a hierarchically organized society where women are, unfortunately, expected to adopt a submissive position for the benefit of the family [10,23]. One study [10] observed enabling behaviors in women, such as providing money or assets for the purchase of psychoactive substances, threatening to break up or separate without the intention of going through with it, and letting the user stay in their home to avoid being homeless or incarcerated. The behavior of male family members is more related to using psychoactive substances together with the user and providing the substance [10]. Holmilia (1994) highlights the need to take care to not transfer the responsibility of drug use to female AFMs when identifying behaviors that facilitate and enable use. In other words, we need to be careful not to blame female caregivers for male users’ use of psychoactive substances [9,23].

This study has limitations, the first of which is related to the sample. There is an imbalance in the gender representation of family members and substance users. In the first case, we have a significantly higher number of women, and in the second case, a significantly higher number of men. Several studies show that this is a reality in the context of substance use, both in terms of female family members seeking more help and in terms of users being more represented by men, or at least, men often being the ones seeking treatment. This is an important point since it is fundamental to consider that this is a sample collected via telephone, in a free service where demand was spontaneous. Another relevant point is that the results of the analyses of the factor structures and the internal consistency were some of the limits of what is acceptable. Therefore, it is essential that in any context in which this instrument is used, new analyses in this regard are carried out and comparisons with this study are made. Furthermore, online applications should also be a point of care, and comparison with a face-to-face sample can greatly contribute to the improvement of this tool.

Future perspectives should include the search, whenever possible, for a sample balance, although the reality of this healthcare context hardly provides opportunities for this. Even so, an effort related to this point should be considered in future studies. Regarding the analysis of the psychometrics of this instrument, it is important to conduct further research in this regard, including in the face-to-face context. This will allow for more robust analyses and pertinent comparisons. To assess the consistency of the results over time and the stability of the construct, it is essential to conduct studies that consider test-retest reliability as much as possible. In addition, considering the identification of other constructs can support the refinement of the instrument. Even so, this work is an initial starting point for the use of this tool. The instrument is the only scale known to date that assesses the frequency of enabling behaviors in AFMs. We suggest that future studies apply this version of the instrument in face-to-face care, to assess whether there is variability in the frequency and manifestation of descriptions of enabling behaviors.

## 6. Conclusions

This study demonstrates that the Brazilian version of the BES has good psychometric properties. Furthermore, the scale’s applicability was extended to all family members of psychoactive substance users, and the analyses resulted in a smaller version consisting of 15 items. The scale is an essential tool for evaluating the permissive behaviors manifested by family members, and it may help individuals to recognize and become psychoeducated regarding enabling behaviors that are neither reasonable nor in the best interest of the drug user.

## Figures and Tables

**Figure 1 ijerph-21-01230-f001:**
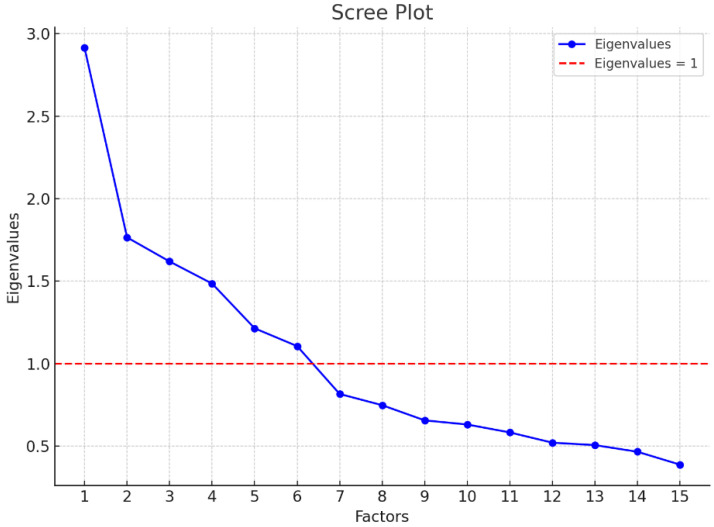
Scree plot graph for factor extraction.

**Figure 2 ijerph-21-01230-f002:**
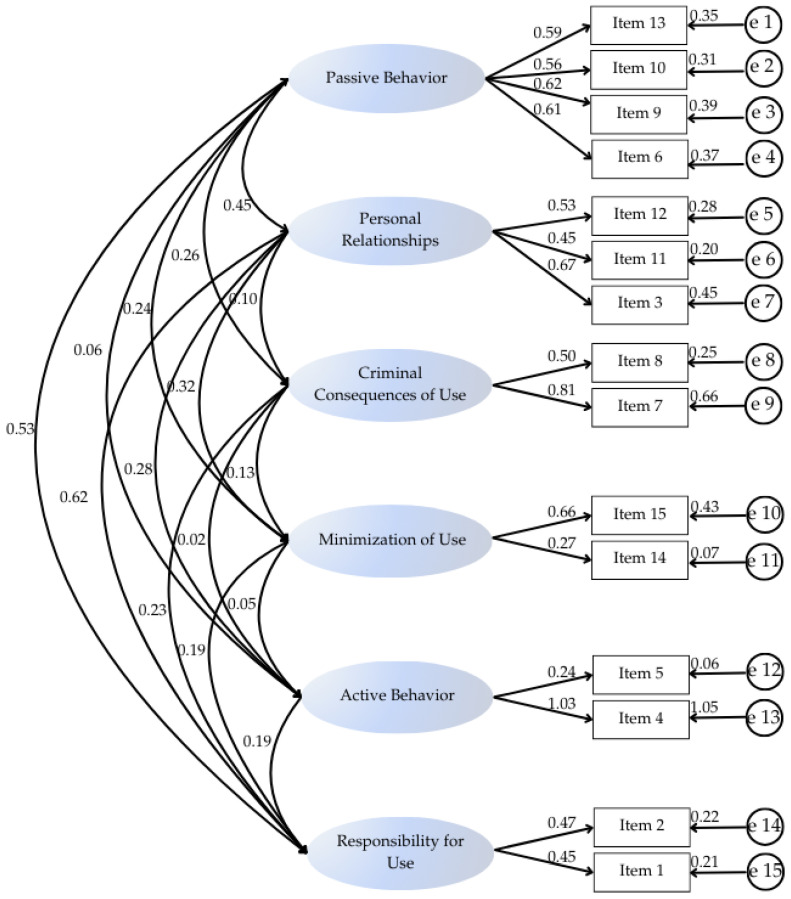
Structural Equation Model of Behavioral Factors and their Associated Items with Factor Loadings.

**Table 1 ijerph-21-01230-t001:** The sociodemographic characteristics of the participants *(n* = 400).

Characteristics of Samples	EFA (*n* = 200) *n*	%	CFA (*n* = 200) *n*	%	*p*
Degree of relatedness					0.016 ξ
Mother	104	52.0	130	65.0	
Father	16	8.0	6	3.0	
Spouse	41	20.5	44	22.0	
Sibling	21	10.5	12	6.0	
Son/daughter	8	4.0	3	1.5	
Other	10	5.0	5	2.5	
Sex of family member					<0.001 §
Female	174	87.0	192	96.0	
Male	26	13.0	8	4.0	
Family member age **¶**					0.731 §
≥45 years old	115	59.6	114	57.9	
<45 years old	78	40.4	83	42.1	
Marital status					0.328 ξ
Married	118	60.2	112	56.3	
Separated/divorced	19	9.7	31	15.6	
Single	45	23.0	40	20.1	
Widowed	14	7.1	16	8.0	
Family income					0.741 ξ
From 1 to 4 minimum wages	110	58.5	119	60.4	
From 5 to 10 minimum wages	56	29.8	57	28.9	
More than 10 minimum wages	22	11.7	21	10.7	
User age					0.266 §
≥25 years old	107	55.7	117	59.7	
<25 years old	85	44.3	79	40.3	
Sex of Substance User					0.758 §
Male	175	87.5	177	88.5	
Female	25	12.5	23	11.5	

§: Pearson’s chi-square test with continuity correction. ξ: Pearson’s chi-square test. ¶: missing data—family member’s marital status [5 (1.3%)]; family income [15 (3.8%)]; education, age, age group of family member [10 (2.5%)]; user age and age group [12 (3.0%)].

**Table 2 ijerph-21-01230-t002:** The structures of the six factors of the BES (Brazilian version) (*n* = 200).

	Commonalities	Factor Loadings
Factor 1 (Passive Behavior), eigenvalue = 2.91%, variance = 15.73%, accumulated variance = 15.73%, ω = 0.70, mean (SD) = 2.2 (1.0).		
6—You have postponed or canceled family meetings or social activities because your (family member) was using (drugs) or had a “hangover”.	0.620	0.666
9—You have helped to take care of your (family member) during a “hangover”.	0.607	0.690
10—You have cleaned up (vomit, urine, etc.) after your (family member) felt sick.	0.622	0.696
13—You have apologized to others for the inappropriate behavior of your (family member) when he/she was under the influence of (drugs).	0.485	0.567
Factor 2 (Personal Relationships), eigenvalue = 1.76%, variance = 11.88%, accumulated variance = 27.61%, ω = 0.65, mean (SD) = 1.7 (0.9).		
3—You have lied or made excuses to family or friends to hide your (family member’s) drug use.	0.572	0.648
11—You have asked or encouraged your family members to ignore or not comment on your (family member’s) use of (drugs).	0.525	0.592
12—You have helped to hide the use of (drugs) by your (family member) from your managers and coworkers.	0.604	0.679
Factor 3 (Criminal Consequences of Use), eigenvalue = 1.61%, variance = 9.84%, accumulated variance = 37.45%, ω = 0.71, mean (SD) = 1.4 (0.7).		
7—You have turned to the police, judge, attorney, or any other professionals, to take your (family member) out of any problem caused by using (drugs).	0.732	0.769
8—You have paid an attorney or bailed your (family member) out due to (drug) problems.	0.721	0.775
Factor 4 (Minimization of Use), eigenvalue = 1.48%, variance = 8.62%, accumulated variance = 46.07%, ω = 0.70, mean (SD) = 1.1 (0.4).		
14—You have reaffirmed to your (family member) that his/her use of (drugs) was not that bad.	0.719	0.749
15—You have lied or told half-truths to a doctor, judge, attorney, counselor, or police officer about your family member’s use of (drugs) or his/her participation in rehab programs.	0.134	0.713
Factor 5 (Active Behavior), eigenvalue = 1.31%, variance = 7.46%, accumulated variance = 53.54%, ω = 0.698, mean (SD) = 1.3 (0.6).		
4—You have used (drugs) together with your (family member) or in his/her presence.	0.542	0.565
5—You have told your (family member) that it is okay to use (drugs) on a few days, on special family occasions, or at social events.	0.720	0.729
Factor 6 (Responsibility for Use), eigenvalue = 1.10%, variance = 6.77%, accumulated variance = 60.31%, ω = 0.627, mean (SD) = 2.0 (1.1).		
1—You have given money to your (family member) to buy (drugs).	0.564	0.551
2—You have taken over tasks from your (family member) because he/she was using (drugs).	0.658	0.711

ω = McDonald’s omega.

**Table 3 ijerph-21-01230-t003:** The goodness-of-fit indices for the six-factor model.

Suitability Parameters	Confirmatory Factorial Model—6 Factors
Global adjustment	
χ^2^/g.l. (<5.0)	2.655
Absolute fit index	
RMR (SRMR)	0.63 (0.038)
RMSEA	0.063 (IC95%: 0.046–0.083)
GFI	0.917
Incremental fit index	
TLI	0.987
NFI	0.916
AGFI	0.961
CFI	0.975

χ^2^/g.l.—Rate between χ^2^ and degrees of freedom (reference value < 5.0). SRMR—Standardized residual mean square root (reference value close to zeroGFI: Goodness-of-fit index (reference value > 0.80). RMSEA: Root mean square error of approximation (reference value between 0.05 and 0.08). NFI—Normed Fit Index (reference value NFI > 0.90). AGFI—Adjusted Goodness-of-Fit Index (reference value AGFI > 0.90). CFI: Comparative fit index. TLI: Tucker–Lewis Index (reference value > 0.90).

## Data Availability

The authors made the data publicly available on the University’s open data portal (https://dados.ufcspa.edu.br/group/institucional).

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
