# Peer review of "The Translated and Adapted Brazilian Version of the Behavioral Enabling Scale for Family Members of Psychoactive Substance Users: An Analysis of the Factorial Structure and Internal Consistency"

_ijerph, 2024, doi:10.3390/ijerph21091230_

Round 1

Reviewer 1 Report (Previous Reviewer 1)

Comments and Suggestions for Authors

Dear Authors

Thank you for considering and accepting my recommendations. In my opinion these revisions have greatly enhanced the quality of the paper.

Author Response

Reviewer 2 Report (New Reviewer)

Comments and Suggestions for Authors

In this study, the researchers conducted both an EFA and a CFA analyses on a translated scale for Behavioral Enabling conducted on a quite large sample of AFM of psychoactive substance users. While the data is kinda interesting, the paper lacks in both formatting/clarity (how you present your results) and scientific robustness (how you support your hypothesis/aim). In this case, the manuscript seems to me not well-written, with many errors, inconclusive sentences, and obscure sentences too. It seems like the manuscript has been written without the necessary care and time to make it suitable to publication, or just comprehension by a potential reader.

About the contents, while some interesting analyses have been conducted, some more are missing. The EFA should be informed by preliminary analyses which are missing (eigenvalues, scree plot, etc...), validation process includes usually some measures of performance, such as convergent validity (correlation with similar scale) or criterion validity (capability to predict or relate to some outcomes). Even methods and results lack of more accuracy in refining the report, and it seems more like a first-stage draft of a manuscript than a paper ready for publication, or even close to this goal.

Overall, my evaluation is not that bad as it can appear by those words, but I really need to see a very different manuscript in a revised version to be convinced to give my positive evaluation for publication.

Here a list of punctual revisions:

ABSTRACT:

This sentence is repeated:

The 25 KMO= 0.680 and Bartlett's test of sphericity= X2 (gl=153) 497.201, p<0.0001 showed significance. 

INTRO:

row 36: the ref is wrong as it start with [14]. Please revise all the refs numbering.

row 80: please report the names of the researcher involved in this research, as it cannot be inferred easily by the ref number [18]. This suggestion applies to all the manuscript.

row 98: "To date, no studies have been found that have used psychometric analysis of the Enabling Behavior Scale in any culture" this is intrisecally false, as the scale has been already applied to a study. Then, you can eventually say that it has not been used in different cultural contexts respect to the original one in which it has been developed.

row 100: again, problem with refs "Rotunda, et al. (2004)" is not in the right format.

METHODS:

row 118: "All consultations were supervised by graduate students in the health area". Have there been any supervision also by experienced researchers or professors? 

row 133: "2.4. Family Care Protocol" this and the following sub-para should be under the "2.3. Instruments" part, not as indipendent ones.

row 171: please describe how you selected or randomized participants for the EFA and CFA. Maybe, some explanations of the sample size estimation could be added also to the participants section.

RESULTS:

Table 1: the table is wrongly displayed. Please provide a better one. Also, please provide a more comprehensive description of user and family memebers: for example, sex is reported only for users.

row 200: "Regarding information from family members" these data are not reported in the table. 

row 203: "This result corroborated the significant difference achieved when comparing the sex of the family member when comparing the two samples (p<0.001)" data about sex is not reported in the table. Moreover, the sentence "so that the female sex, 96.0% (n=192), was associated with the CFA sample. In contrast, the male sex was related to the EFA sample, 13.0% (n=26)." is no-sense: what does it mean "was associated with"? Maybe, there are more females in the CFA sample than in the EFA sample, and viceversa for males. Is this correct? Please clarify this sentence.

row 224: the original scale has 20 items. You reported the exclusion of Items 2, 7, and 17, then the final scale has, for no reasons, 15 items. 20-3 = 17. Where are the two missing items? 

Table 2: please provide the factor loadings and statistics for all the tested items before the exclusion of the selected items. Moreover, please provide fit statistics before and after the removal of such items, and compare the two models.

row 229: in general, prior to conduct an EFA, is a good practice to conduct some screening about the number of factors to be extracted, by means of a scree plot and/or eigenvalues. Please provide such analysis to justify the use of a six-factor model.

row 230: 60,318%% correct this.

row 249: how could it be "r=0.053" the highest value in correlation?

row 250: this sentence "Also, having a Bartlett’s sphericity test of , p <0.0001; several criterion of commonal ities with values above 0.500" is not understandable. Please revise it.

row 255: conducting Cronbach's alpha evaluation on two-scale items is problematic, as it cannot be inferred correctly. Moreover, if you consider a total value, please also test with the CFA a bifactor model, including a general factor summarising the overall construct.

DISCUSSION:

In general, please remove all the statistics (such as "Bartlett's sphericity test also showed statistical significance") from the discussion. They were already reported in the Results section and should not be repeated here.

row 312: "The small number of men causes curiosity among researchers" this sentence has no sense. This is maybe a limitation of the study, but this is also not fixable as it depends on the characteristics of the familiars who accesses the services (and of the substance users).

row 332: another pointless sentence: "The present study also has a limitation related to the significantly more significant number of women than men"

Comments on the Quality of English Language

The quality of the English should be improved. Please take care of this side of your manuscript preparation.

Round 2

Reviewer 2 Report (New Reviewer)

Comments and Suggestions for Authors

Dear Authors,

thanks for your work on this manuscript. You answered all my questions, and I am happy to see many more details in the text now. However, your manuscript again needs a profound editing, as it somewhat confused and includes a number of typos or errors (for example, row 177 "Factor analysis was conducted the modified scale" or row 215-16 "male sex [EFA: 215 87.5% (n=175) vs. CFA: 88.5% (n=177)] dominated"?). Please, proofread carefully your text with the help of a professional proofreader or by an AI tool, it will very much increase the readability of your manuscript.

Then, the methods and results are clearer than before. Again, however, there is room for improvements. First, please report in text (not only to me!) your preliminary analysis for the EFA, showing the eigenvalues etc. This is crucial for readers for trusting your words. Second, please carefully rewrite the Statistical analysis section, which is messy at the present moment. Third, the tables should be presented in a coherent format across tables, now it seems like you copy-paste tables from different software into the manuscript. Please have care for these important details! 

Lastly, in the discussion please add a compelling Limitations section, with a list of potential limitations of the study (starting from the unbalanced sample) and presenting a full recognition of all the issues at hand. Conclude this section by proposing amendments for future works.

Comments on the Quality of English Language

See the general comments.

Round 3

Reviewer 2 Report (New Reviewer)

Comments and Suggestions for Authors

Dear Authors,

you did a nice job in responding to all my concerns. I am fully supportive of publishing your manuscript. A last suggestion before publication, please increase the quality and readability of your figures, and use only the second panel on Figure 1 as is the only one relevant for your models.

Congratulations!

This manuscript is a resubmission of an earlier submission. The following is a list of the peer review reports and author responses from that submission.

Round 1

Reviewer 1 Report

Comments and Suggestions for Authors

Thank you for inviting me to review this publication where an existing scale has been adapted for use in Brazil. The paper is worthy of publication with some minor amendments as follows:

1. The current title implies that this is the first time the BES has been assessed for psychometric properties. The title needs to reflect the translation, cultural adaptation and modification of an existing scale from English to Portuguese.

2. Please remove Portuguese text from the front of the paper.

3. Add the title - Introduction

4. The introduction is clear and informative. Please refer to the current EBS being in English and possibly refer to the country in which it was developed. This helps to provide a background to the need for a cultural adaption of the scale as well as simply a translation with psychometric assessment.

5. Participants - 2nd parag o remove first completely in 'completely completed'.

6. 2.5 - Brazilian version - refer to any translation and back translation practices that were undertaken in line with standard approaches to this. Also refer to any cultural adaption that was identified during this process, so that the adaptation was not simply addition of new items / content.

7. Table 1 - family income. from 1 to 5 or from 5 to 10. What happens if the response is 5 - adjust accordingly.

8. Were Cronbach's alpha values checked for the individual scales or only the overall scale value? Please add individual scale alphas.

Author Response

Thank you, for your review and comments for the improvement of the manuscript. We tried our best to follow your suggestions. 

  1. The current title implies that this is the first time the BES has been assessed for psychometric properties. The title needs to reflect the translation, cultural adaptation and modification of an existing scale from English to Portuguese.

We modified the title to: “Psychometric analysis of the translated and adapted Brazilian version of the Behavioral Enabling Scale  of family members of psychoactive substance users”

  1. Please remove Portuguese text from the front of the paper.

Suggestion accepted - The abstract in Portuguese was excluded from the beginning of the manuscript.

  1. Add the title - Introduction

Suggestion accepted - Title was inserted at the beginning of the section

  1. The introduction is clear and informative. Please refer to the current EBS being in English and possibly refer to the country in which it was developed. This helps to provide a background to the need for a cultural adaption of the scale as well as simply a translation with psychometric assessment.

Suggestion accepted - Specific information about the place where the scale was developed was inserted in the fourth paragraph of the introduction.

  1. Participants - 2nd paragraph to remove first completely in 'completely completed'.

Suggestion accepted - word has been removed. 

  1. 2.5 - Brazilian version - refer to any translation and back translation practices that were undertaken in line with standard approaches to this. Also refer to any cultural adaptation that was identified during this process, so that the adaptation was not simply addition of new items / content.

Suggestion accepted- We included a paragraph to describe the methods in the item “2.5. Behavioral Enabling Scale (Brazilian version)”

  1. Table 1 - family income. from 1 to 5 or from 5 to 10. What happens if the response is 5 - adjust accordingly.

Suggestion accepted - Values were adjusted as suggested.

  1. Were Cronbach's alpha values checked for the individual scales or only the overall scale value? Please add individual scale alphas.

In table 2 the readers can verify the Eigenvalue, % variance, % accumulated variance, Cronbach’s Alpha, Mean (SD) for each factor, in bold.

Reviewer 2 Report

Comments and Suggestions for Authors

This paper aims at validating the Behavioral Enabling Scale Brazilian, as a valid tool that measures the frequency of enabling behaviors of family members of psychoactive substance users. The article is well-written and structured. However, some revisions are needed.

 Abstract

I appreciate that the authors have included the abstract also in their native language. However, it should be structured like the one in English. Moreover, please introduce a space between Palavras-chave and the introduction of the paper. Moreover, add the title of the section “1. Introduction”.

 Please, check the objective of the study described in the abstract. The authors must underline that they want to test the translated and adapted Brazilian version of the Behavioral Enabling scale.  [also in the title I would underline the version you are using]

Introduction

The introduction is complete and full of details. Please, add references for all concepts.

Methods

1] who translated and adapted the scale?

2] How did you check that the items were understandable by users?

3] Was the factor analysis conducted using the "modified" scale? Please, detail this aspect.

Results

1] Please, clearly state which items were excluded

2] Table 2 is a bit confusing. I suggest the authors divide it into two tables.

3] “Regarding the results found in the bivariate analysis, there are no significant differences for sociodemographic characteristics” Please, add p-values.

4] According to the authors, is the sample enough? The authors could do a post-hoc sample size calculation to evaluate the statistical power of the study.

Discussion

The authors could more detail the clinical utility of this validation.

 Other point

Please, make an English check (grammar and spelling).

Author Response

Thank you, for your review and comments for the improvement of the manuscript. We tried our best to follow your suggestions and included a response for the topics that were suggested by the reviewer.

Introduction

The introduction is complete and full of details. Please, add references for all concepts. 

Response:  Please consider that the multiple citations after a group of phrases are referring to the previous concept.

Methods

1] Who translated and adapted the scale?

Please see the item “2.5. Behavioral Enabling Scale (Brazilian version)”: the last paragraph was included in this new version of the manuscript. 

2] How did you check that the items were understandable by users? The translated scale was applied to a few callers  to the hotline that accepted to give their opinion about the practical understanding of the questions.

Please see the last sentence in the item “2.5. Behavioral Enabling Scale (Brazilian version)”.

3] Was the factor analysis conducted using the "modified" scale? Please, detail this aspect. Suggestion accepted. Please see the sentence in the 3rd parag of the statistical sector:   “Factor analysis was conducted using the modified scale, with no questions about the spouses “

Results

1] Please, clearly state which items were excluded.

Suggestion accepted, please see in page 7 of 12: “...., 3 items were excluded (Items 2, 7, and 17 in the original scale by Rotunda et al. 2004.

2] Table 2 is a bit confusing. I suggest the authors divide it into two tables.

We agree that table 2 was badly formatted. Because it would be odd to separate the 6 actors into different tables we decided to maintain Table 2 as with all Factors included and gave more space between the description of each factor. 

3] “Regarding the results found in the bivariate analysis, there are no significant differences for sociodemographic characteristics” Please, add p-values. Suggestion accepted and the p value >0.05 was included in Table 1

4] According to the authors, is the sample enough? The authors could do a post-hoc sample size calculation to evaluate the statistical power of the study.

Response: Yes, we used the criteria of having 10 participants included in the study for each question in the scale, as in Pasquali (Pasquali L. Instrumentação psicológica: fundamentos e prática. Porto Alegre: Editora Artmed; 2010.). Also, having a Bartlett’s sphericity test of , p <0.0001; several criterion of commonalities with values ​​above 0.500, and quite high  Cronbach coefficient show satisfactory internal  validity of the study. 

Discussion

The authors could more detail the clinical utility of this validation. 

Suggestion accepted, information was added in the manuscript in the first paragraph of the discussion, and also in the last sentence of the conclusion

 Other point

Please, make an English check (grammar and spelling).  

English language check was performed, and  adjustments were made in the new manuscript.

Reviewer 3 Report

Comments and Suggestions for Authors

The revised work is conveniently presented. It is considered of interest for the scientific community as well as in the clinical field, for the intervention and prevention of relapse in the field of addictions. This initiative is appreciated.

Regarding the introduction, the proposed theme is addressed following a clear common thread, with antecedents, relevant aspects (family system, behaviors that facilitate consumption) as well as the population to be analyzed, men and women of the general population who consume psychoactive substances.

In addition, the theoretical framework of analysis is contextualized, with updated references and authors/researchers specialized in the subject matter under study are indicated, so that knowledge of the subject is revealed.

The results are presented clearly, concisely and satisfactorily respond to the objectives of the study. Both in formal aspects and in the selection of the design and selected tests are adequate. The sequence of paragraphs is coherent, appropriately responding to the object of study. The tables and figures are perfectly presented, both in formal aspect and in content (adequate) clear and adjusted to the regulations (except in the numerical values, which it is suggested to review. For example, in the values in % three decimal places are included).

Both the discussion and the conclusions, as well as the limitations of the study, are clearly stated.

However, throughout this section, some deficiencies are pointed out:

a) The title does not accurately reflect the study object or population, that is, consumption of psychoactive substances is indicated, but in reality only participants with alcohol consumption are included. This aspect generates confusion throughout the manuscript (not only in the title and abstract), without making this differentiation clear when referring to previous studies. Therefore, it is considered necessary to review this aspect throughout the manuscript, that is, to specify when reference is made to the population with alcohol consumption and when other types of consumers are included.

b) Information is included on the consequences of the presence of a consumer in the family. However, its purpose is not clear in a study that aims to evaluate the psychometric measures of an instrument. Any reference to this is missing in the final section of the manuscript, or at least that it is clear why the authors consider it relevant to include it in this type of manuscript.

c) In relation to the instrument, developed in 2004, it is proposed that translation and adaptation be carried out, but throughout the work the relevant aspects necessary in this process are not specified. Likewise, there is no justification (throughout the manuscript) for why some of the items (two) are eliminated, instead of reformulating them to adapt them to more family members, it is not indicated which they are, nor are they indicated. the three that disappear from the initial version of 20 to the final of 15 items. It is considered important to record this information, as well as to clarify in the text the non-coincidence in the numbering of the items, to facilitate reading comprehension (although they do appear in Figure 1).

d) Likewise, given the role of family members as possible risk factors, it is considered important to carry out previous studies that show whether there are differences between those families in which it is consumed, compared to those in which it is consumed. those that are not consumed, specifically alcohol.

d) Regarding citations and bibliographical references, it is recommended, on the one hand, that they comply with the required criteria (according to the journal, either by alphabetical order, year of publication, etc.), that is, that a criterion be followed and , on the other hand, the bibliography is updated with studies after 2021, especially in the population under study.

e) In the statistical analysis section, reference is made to Bortolon (2015), as a previous study of the instrument. It is recommended that the most relevant aspects of this author be included, as well as that it be clarified if the current instrument is an adaptation or modification. Confusion is generated in the manuscript.

f) Regarding Informed Consent, how was it carried out? Written or verbal. And the completion of the instrument?

In conclusion, the objective and development of the study is positively valued. However, some modifications are proposed for the improvement and quality of the manuscript.

Author Response

Thank you, for your review and comments for the improvement of the manuscript. 

  1. a) The title does not accurately reflect the study object or population, that is, consumption of psychoactive substances is indicated, but in reality only participants with alcohol consumption are included. This aspect generates confusion throughout the manuscript (not only in the title and abstract), without making this differentiation clear when referring to previous studies. Therefore, it is considered necessary to review this aspect throughout the manuscript, that is, to specify when reference is made to the population with alcohol consumption and when other types of consumers are included. 

Please see sector 2.2 where we state “We analyzed 200 protocols of family members of psychoactive substance users who called the Ligue 132 hotline service, from the National Service for Guidance and Information on Drug Use”. Family members themselves would use tobacco or alcohol, but their family members were users of psychoactive substances in general, alcohol and illicit drugs. The original scale developed by Rotunda was applied to spouses of users of alcohol only, in our study we extended the application to all substances and we adapted and validated the scale for drug use in general (except tobacco).

As suggested by the other reviewers we modified the title of the manuscript to: “Psychometric analysis of the translated and adapted Brazilian version of the Behavioral Enabling Scale  of family members of psychoactive substance users”

  1. b) Information is included on the consequences of the presence of a consumer in the family. However, its purpose is not clear in a study that aims to evaluate the psychometric measures of an instrument. Any reference to this is missing in the final section of the manuscript, or at least that it is clear why the authors consider it relevant to include it in this type of manuscript. To clarify this question we included a new sentence referring to Affected Family Members. 
  2. c) In relation to the instrument, developed in 2004, it is proposed that translation and adaptation be carried out, but throughout the work the relevant aspects necessary in this process are not specified. Likewise, there is no justification (throughout the manuscript) for why some of the items (two) are eliminated, instead of reformulating them to adapt them to more family members, it is not indicated which they are, nor are they indicated. the three that disappear from the initial version of 20 to the final of 15 items. It is considered important to record this information, as well as to clarify in the text the non-coincidence in the numbering of the items, to facilitate reading comprehension (although they do appear in Figure 1).

Response: The two items that were specific to spouses were not adapted to others because they were very specific to a partner role. Item 9 in Rotunda: Mr. (a) had an intimate relationship with (relative), when she did not want to, because he (she) had used (drugs). Item 11- Mr. (a) threatened to separate your (family member) due to the use of (drugs) but then did not separate.Regarding the other excluded items, suggestion accepted, please see in page 7 of 12: “...., 3 items were excluded (Items 2, 7, and 17 in the original scale by Rotunda et al. 2004). Also, we accepted the suggestion of all reviewers to expand the explanation on translation and adaptation in a paragraph to describe the methods in the item “2.5. Behavioral Enabling Scale (Brazilian version)”: the last paragraph was included in this new version of the manuscript. 

  1. d) Likewise, given the role of family members as possible risk factors, it is considered important to carry out previous studies that show whether there are differences between those families in which it is consumed, compared to those in which it is consumed. those that are not consumed, specifically alcohol. 

Response: In fact this idea is quite interesting, but very difficult to carry on in Brazil. Alcohol use is very prevalent among our population, including  young adolescents (https://cisa.org.br/biblioteca/downloads/artigo/item/356-panorama2022). Also, in this study, we studied  a population of family members of consumers of all licit and illicit psychoactive substances and polyusers, and it would be impossible to compare users of each combination of drugs with no users for the validations of the scale. 

  1. d) Regarding citations and bibliographical references, it is recommended, on the one hand, that they comply with the required criteria (according to the journal, either by alphabetical order, year of publication, etc.), that is, that a criterion be followed and , on the other hand, the bibliography is updated with studies after 2021, especially in the population under study.

Suggestion accepted, references from the year 2023 were added to the manuscript. 

  1. e) In the statistical analysis section, reference is made to Bortolon (2015), as a previous study of the instrument. It is recommended that the most relevant aspects of this author be included, as well as that it be clarified if the current instrument is an adaptation or modification. Confusion is generated in the manuscript. 

Response: we agree that the text got confusing an cited Bortolon 2015 for the information on comprehension of the questions by the Family Members.

  1. f) Regarding Informed Consent, how was it carried out? Written or verbal. And the completion of the instrument?

Response: The sentence: “ The informed consent and the completion of all instruments were done verbally through the proactive telephone call made by the family member to the hotline. “ was included in the second paragraph of the 2.2 -Participants sector of the manuscript.